# Non-Small Cell Lung Cancer Patients with Skip-N2 Metastases Have Similar Survival to N1 Patients—A Multicenter Analysis

**DOI:** 10.3390/jpm15030113

**Published:** 2025-03-14

**Authors:** Georg Schlachtenberger, Simon Schallenberg, Fabian Doerr, Hruy Menghesha, Christopher Gaisendrees, Andres Amorin, Alberto Lopez-Pastorini, Reinhard Büettner, Alexander Quaas, David Horst, Frederick Klauschen, Nikolaj Frost, Jens C. Rueckert, Jens Neudecker, Khosro Hekmat, Matthias B. Heldwein

**Affiliations:** 1Department of Thoracic Surgery, Hildegardis Hospital Cologne, Bachemer Strasse 29–33, 50931 Cologne, Germany; 2Department of General, Visceral and Thoracic Surgery, University Hospital of Cologne, 50937 Cologne, Germany; andres.amorin-estremadoyro@uk-koeln.de (A.A.); alberto.lopez-pastorini@uk-koeln.de (A.L.-P.); khosro.hekmat@uk-koeln.de (K.H.); matthias.heldwein@uk-koeln.de (M.B.H.); 3Department of Pathology, Charité—Universitätsmedizin Berlin, 10117 Berlin, Germanydavid.horst@charite.de (D.H.); frederick.klauschen@med.uni-muenchen.de (F.K.); 4Department of Thoracic Surgery, University Medicine Essen—Ruhrlandklinik, University Duisburg-Essen, 47057 Duisburg, Germany; 5Department of Thoracic Surgery, Helios Clinic Bonn/Rhein-Sieg Bonn, 53123 Bonn, Germany; 6Department of Cardiac Surgery, University Hospital of Cologne, 50937 Cologne, Germany; christopher.gaisendrees@uk-koeln.de; 7Department of Pathology, University Hospital of Cologne, 50937 Cologne, Germany; reinhard.buettner@uk-koeln.de (R.B.); alexander.quaas@uk-koeln.de (A.Q.); 8Department of Infectious Diseases and Respiratory Medicine, Charité—Universitätsmedizin Berlin, 10117 Berlin, Germany; nikolaj.frost@charite.de; 9Department of Surgery, Charité—Universitätsmedizin Berlin, 10117 Berlin, Germany; jens-c.rueckert@charite.de (J.C.R.); jens.neudecker@charite.de (J.N.)

**Keywords:** non-small cell lung cancer, lymph nodes, skip-N2 metastases, N1 lymph nodes

## Abstract

**Introduction:** Nodal involvement is one of the most important prognostic factors in NSCLC. Skip-N2 metastasis (N0N2), which is N2 metastasis in the absence of N1 metastasis, occurs in approximately 20–30% of patients. According to the International Association for the Study of Lung Cancer, N1 and N0N2 patients may have comparable long-term survival, considering their similar tumor stages. However, this conclusion remains controversial. Therefore, we carried out this multicenter study to examine the long-term survival and disease-free interval (DFI) of N0N2- and N1 patients. **Methods:** One-, three-, and five-year survival rates were measured. Kaplan–Meier curves and a Cox proportional hazards model assessed survival and were used to identify prognostic factors for overall survival. **Results:** Between January 2010 and December 2020, 273 N0N2 and N1 patients were included in our analysis. Of those patients, 77 showed N0N2 and 196 N1. Baseline characteristics did not differ significantly between groups. Between N0N2 and N1 patients, there were no significant differences in one- (*p* = 0.67), three- (*p* = 0.20), and five-year (*p* = 0.27) survival. Furthermore, DFI did not differ between groups (*p* = 0.45). **Conclusions:** Our findings indicate that N0N2 patients have a prognosis comparable to that of patients with N1 disease. These results indicate that patients with N0N2 have a similar prognosis to N1 patients. N2-NSCLC is heterogeneous and would benefit from a more precise subdivision and differential treatment in the upcoming UICC 9 classification. The following question remains: are we overtreating N0N2 patients or undertreating N1 patients?

## 1. Introduction

Lung cancer remains one of the leading causes of cancer-related mortality, with an annual incidence exceeding two million cases worldwide [1]. Surgical resection is the standard treatment for early-stage non-small cell lung cancer (NSCLC). However, the optimal therapeutic approach for locally advanced NSCLC remains controversial [2]. At diagnosis, approximately 30% of NSCLC patients present with a locally advanced disease, often characterized by lymph node involvement [3,4]. Nodal involvement is one of the most important prognostic factors in NSCLC. If a mediastinal lymph node involvement is evident, a multimodal therapeutic approach consists of upfront surgery plus adjuvant therapy or neoadjuvant therapy plus surgery [2]. In the presence of hilar involvement only, surgery remains the first therapeutic option with adjuvant treatments [5]. However, N2 disease includes a large spectrum of mediastinal involvement, ranging from single metastatic lymph nodes to bulky disease. Approximately 30% of N2 metastases occur without prior N1 involvement, a phenomenon known as skip-N2 metastasis (N0N2) [4,6,7,8,9]. In contrast, continuous-N2 metastasis (N1N2) follows a sequential pattern, with both N1 and N2 lymph nodes being progressively affected.

The International Association for the Study of Lung Cancer (IASLC) proposed a sub-classification for the N descriptors, including N0N2 [10]. The proposal categorized N1 patients with hilar involvement if single or multiple stations were involved and with N2 single station involvement without or with N1N2. The IASLC depicted that N1 and N0N2 patients may have comparable long-term survival. They considered that the tumor stage was similar, and therefore, both groups benefited from the same therapeutic approach [5,11].

We carried out this multicenter study to examine if patients with N0N2 and N1 have significant prognostic differences in long-term survival or if the two groups have comparable survival outcomes. We compared the disease-free interval (DFI) and the survival of patients with N0N2 with N1.

## 2. Materials and Methods

This multicenter retrospective cohort study included patients from three thoracic surgical centers, and two pathological centers analyzed specimens.

N0N2 is associated with direct lymphatic drainage into the mediastinal nodes, bypassing the typical spread through pulmonary and hilar lymph nodes.

N1 is only associated with direct lymphatic drainage into the pulmonary and hilar lymph nodes.

### 2.1. Ethics Statement

This study was conducted according to the guidelines of the Declaration of Helsinki and approved by the Ethics Committee of University Hospital of Cologne and Charité—Universitätsmedizin Berlin (protocol code [UK01102021GS] and date of approval [18 January 2021]). Individual informed consent was waived due to the retrospective observational nature of the study.

### 2.2. Inclusion and Exclusion Criteria

Between January 2010 and December 2020, patients with stages N1 and N0N2 were included in our analysis.

Only patients who underwent an anatomical R0 resection were included. We excluded those with incomplete lymphadenectomy, small cell lung cancer, or perioperative mortality within 30 days.

### 2.3. Diagnostics and Therapy

According to clinical guidelines, patients with suspected lung cancer underwent staging and diagnostic evaluation [2]. Each case was reviewed by a multidisciplinary tumor board (MDT) to determine the optimal management approach.

Diagnostic and staging procedures included high-resolution computed tomography (CT), cerebral magnetic resonance imaging (cMRI), and positron emission tomography-CT (PET-CT). All patients underwent endobronchial ultrasound (EBUS) for further lymph node assessment. The preferred surgical approach for anatomical resection was video-assisted thoracic surgery (VATS). If VATS was not feasible—due to factors such as adhesions or bulky disease—thoracotomy was performed instead.

Lymph node dissection included the hilar (station 10), interlobar (station 11), and lobar (station 12) lymph nodes. Segmental (station 13) and subsegmental lymph nodes (station 14) were retained in the specimens. On the left side, the dissection involved the aortic nodes (stations 5 and 6) and the inferior mediastinal lymph nodes (stations 7, 8, and 9). On the right side, the superior mediastinal lymph nodes (stations 2 and 4) and inferior mediastinal lymph nodes were dissected.

Following guidelines during this period, patients with N0N2 and N1 disease received adjuvant chemotherapy consisting of cisplatin, vinorelbine, or a combination of cisplatin and etoposide for four to six cycles. Patients experiencing significant side effects from cisplatin were switched to carboplatin. Additionally, N0N2 (UICC stage IIIA) patients underwent radiation therapy with a total dose of 66 Gy [2].

### 2.4. Statistical Analysis

Follow-up data were collected from all participating lung cancer centers, with survival queries conducted at the end of December 2023. The lymph node ratio (LNR) was analyzed, defined as the proportion of metastatic lymph nodes relative to the total number of lymph nodes resected.

DFI was measured between the time of surgery and the occurrence of a recurrence in any location, confirmed by histology. One-year, three-year, and five-year survival were calculated from surgery to death. Survival differences were analyzed using Pearson’s χ^2^. The median overall survival was assessed using Kaplan–Meier curves and compared using a log-rank test. The Cox regression model was evaluated for multivariable survival analysis described by hazard ratio (HR) and 95% confidence interval (CI). It was conducted for factors significant in the univariate analysis, assuming they were independent factors for overall survival.

Pearson’s χ^2^ test or Fisher’s exact test were utilized to compare categorical data, depending on the minimum expected count in each cross-tabulation. Continuous parameters were expressed as mean ± standard deviation (SD) and analyzed using an unpaired Student’s *t*-test. A *p*-value of less than 0.05 was considered statistically significant.

## 3. Results

Between January 2010 and December 2020, about 1100 NSCLC patients underwent anatomical resection for NSCLC in three thoracic surgical departments.

The mean follow-up duration for the study group was 38.5 (SD: 29.5) months. After the inclusion and exclusion criteria, 273 patients remained for further analysis. Of those patients, 77 showed N0N2 and 196 N1.

### 3.1. Baseline Characteristics

Table 1 presents baseline characteristics. Neither age (*p* = 0.31) nor gender (*p* = 0.12) differed between groups. Smoking history did not differ (smoked pack-years *p* = 0.69).

Most resected specimens were adenocarcinomas (AC) (72.7%). The remaining tumors (28.3%) were squamous cell carcinomas (SCC). The groups had no significant difference (AC *p* = 0.11, SCC *p* = 0.19). Tumor localization did not differ between groups. In both groups, tumors were most frequently diagnosed in the right upper lobe. The proportion of segmentectomies, lobectomies, bilobectomies, and pneumonectomies did not differ between groups. There was no significant difference in the LNR (N0N2: 0.13 (SD: 0.1) vs. N1: 0.15 (SD: 0.2); *p* = 0.61).

Table 2 presents baseline characteristics: grading and lymph nodes. Number of metastatic lymph nodes (N0N2: 3.0 (SD: 3.3). N1: 2.3 (SD: 1.9); *p* = 0.08) as well as tumor size (N0N2: 3.5 (SD: 2.6) vs. N1: 3.9 (SD: 3.4) *p* = 0.27) did not differ significantly between groups. Tumor grading did not differ significantly between groups. Preoperative N-status was accurately determined in only 47% of patients. Specifically, N1 was correctly diagnosed in 65% of cases, compared to just 18% for N0 or N2 (*p* = 0.002).

Patients’ comorbidities were recorded and classified into two groups: one comorbidity versus multiple comorbidities. No significant differences were observed between these groups (*p* = 0.56, *p* = 0.23).

### 3.2. Overall Survival and Disease-Free Survival

The one-year survival rate of N0N2 patients was 90.9% compared to 89.6% of N1 patients (*p* = 0.67). After three years, 65.3% of N0N2 patients were alive compared to 60.3% of N1 patients (*p* = 0.20). After five years, 40.7% of N0N2 patients were still alive compared to 44.7% of N1 patients (*p* = 0.27). We assessed overall survival between groups using the Kaplan–Meier method. Survival curves did not differ significantly between groups. (log-rank *p* = 0.51). Disease-free interval (DFI) did not differ between groups (40.7 (SD: 31.1) vs. 38.1 (SD 36.7); *p* = 0.76).

Table 3 presents one-year, three-year, and five-year survival rates, overall survival rates, and DFI. Survival and curves are shown in Figure 1a,b.

Cox regression analysis indicated that the occurrence of N0N2 versus N1 had no significant impact on overall survival. (HR 1.18, CI 0.79–1.78; *p* = 0.4).

Moreover, tumor size, grading, and metastatic lymph nodes had no significant influence on overall survival in this cohort.

Cox regression analysis is presented in Table 4.

## 4. Discussion

Nodal involvement remains among the most important prognostic factors in NSCLC [11].

However, the N-status alone does not always determine long-term survival.

Therefore, we conducted a multicenter study and collected data from 273 eligible patients with stage N0N2 or N1 disease.

### 4.1. What Are the Reasons for the Occurrence of Skip-N2 Metastases?

The prevalence of N0N2 occurs in approximately 30% of pN2 cases [4]. N0N2 represents cases where bronchopulmonary segments spread directly to mediastinal nodes instead of following the typical pulmonary-to-hilar lymph node route, skipping certain lymphatic pathways.

N0N2 is associated with direct lymphatic drainage into the mediastinal nodes, bypassing the typical spread through pulmonary and hilar lymph nodes. This phenomenon is hypothesized to result from neglect or bypassing intrapulmonary or subpleural lymphatic vessels [12]. The literature suggests N0N2 is more common among older male and smoking patients and is particularly linked to squamous cell carcinoma (SCC) [13]. However, our cohort did not show significant associations between N0N2 and these parameters.

### 4.2. Is N1 Involvement or Skip-N2 Metastases a Prognostic Factor for Overall Survival and the Disease-Free Interval?

This study determined that these two groups of patients, representing two different UICC (Union International Control Cancer) stages in the TNM staging system, did not reveal significant survival differences. The literature has already described a similar prognosis in N0N2 and N1 patients [14,15]. They suggest that patients with N0N2 are similar to N1 patients and less so with patients with continuous-N2 metastases N1N2 [6,7].

Chiappetta et al. analyzed patients based on their lymph node involvement. They found that only the number of affected lymph nodes and the lymph node ratio (LNR) significantly influenced overall survival in adenocarcinoma patients [16]. In contrast, our results show no differences in survival or UICC stage when stratified by histology.

Other studies investigated the prognosis in N1 patients compared to N0N2 patients. Most of those studies did not present definitive results due to a wide range of possible confounding factors, such as the limited number of patients or a variability in the adjuvant therapy [15,17].

In our multicenter analysis, we included 273 N1 and N0N2 patients. To our knowledge, this is one of the biggest cohorts regarding this particular prognostic factor. Our inclusion dates were from 2010 to 2020. At this time, adjuvant therapy differed significantly between groups. N0N2 and N1 patients received cisplatin, vinorelbine or cisplatin, and etoposide as adjuvant chemotherapy for four to six cycles. N0N2 patients underwent mediastinal radiation therapy with a fraction of 66Gy [2]. Since then, guidelines have changed substantially [5,18,19,20].

Following the publication of the Lung ART trial, the approach to adjuvant radiation therapy has evolved significantly. Pechoux et al. demonstrated that adjuvant chemoradiation therapy did not improve disease-free survival [21]. As a result, mediastinal radiation is no longer recommended for stage IIIA NSCLC [19]. Nevertheless, it would have been interesting to determine whether N1 patients who received chemoradiation therapy had better survival outcomes. However, this approach contradicted at least the German guidelines and was therefore not conducted.

For more than 20 years, cisplatin-based chemotherapy was the primary treatment for N1 and N2 NSCLC, with few advancements. Recently, adjuvant immunotherapy has transformed the treatment landscape for these patients. Specifically, monoclonal PD-L1 antibodies such as atezolizumab and nivolumab have significantly improved disease-free intervals (DFI) as adjuvant therapies [22,23,24]. As a result of these excellent outcomes, both therapies have been incorporated into the ESMO guidelines as recommended adjuvant treatment options [5,18,19,20]. Additionally, Osimertinib has been approved for patients with epidermal growth factor receptor mutation-positive NSCLC.

Neoadjuvant therapy for N0N2 and N1 patients is also an option, especially since Nivolumab and Pembrolizumab are approved as neoadjuvant treatments [25].

Neoadjuvant treatment was not approved between our cohort’s inclusion dates, so it was not included in our analysis [26].

Wang et al. confirmed the absence of survival differences comparing N1 and N0N2 patients. Interestingly, the authors performed a sub-analysis of single station N1 vs. multilevel N1 vs. N0N2 patients without significant differences in disease-free survival. Still, in overall survival terms, single station N1 and N0N2 presented a similar prognosis, while multilevel N1 presented a significantly worse outcome (*p* = 0.007) [8,27]. In this subgroup analysis, the group sizes varied greatly. There were 126 patients in single station N1 and 43 in multilevel N1 [4,8,27]. We did not perform a subgroup analysis because there were no significant differences between groups independently of the number of positive lymph nodes (Table 2).

### 4.3. How Should We Treat N0N2 and N1 Patients?

Our multicenter analysis demonstrated comparable overall survival and disease-free intervals for patients with N1 and N0N2 diseases.

Due to the inclusion period between 2010 and 2020, adjuvant treatment differed between groups. Until the end of 2022, stage IIIA patients received chemoradiation therapy, whereas pN1 patients received chemotherapy only. As mentioned above, chemoradiation therapy is no longer the adjuvant therapy for N2 patients, with some minor exceptions [21]. Furthermore, all patients received cisplatin + vinorelbine as adjuvant chemotherapy. Since chemoimmunotherapy and check-point inhibition were included foremost in 2022 in the guidelines.

Another option is neoadjuvant therapy, particularly given the pending EMA approval of Nivolumab for use in the neoadjuvant setting [25].

In the next years, whether the survival curves between N1 and N0N2 will be significantly different through the implementation of immunotherapy as adjuvant and neoadjuvant treatment will be seen.

Further studies are needed to show if survival curves between N1 and N0N2 patients stay similar since adjuvant therapy, according to the current guidelines, was adjusted in 2022. Furthermore, the Ninth Edition of the TNM Classification includes a subclassification of the N2 descriptor. What impact this will have on long-term survival and therapeutic options remains to be seen [28].

### 4.4. Limitations

This retrospective multicenter analysis is subject to the inherent limitations of retrospective research, including reliance on postoperative pathology data. The survival data may not specify the exact causes of death; however, it is most likely that NSCLC was the primary cause.

Additionally, given the period during which the patients were treated, none of the patients in our cohort received immunotherapy or checkpoint inhibitors in the neoadjuvant or adjuvant settings. Further research is needed to evaluate outcomes as these therapies become more widely implemented.

## 5. Conclusions

This multicenter study demonstrated that N0N2 patients had overall survival and disease-free intervals comparable to those observed in N1 patients. Our findings further suggest that the prognosis for patients with N0 or N2 disease is similar to that of patients with N1 involvement. N2 NSCLC is heterogeneous and will benefit from the planned, more precise subdivision in UICC-9 and differentiated treatment. The question remains whether N0N2 patients are overtreated or N1 patients are undertreated.

## Figures and Tables

**Figure 1 jpm-15-00113-f001:**
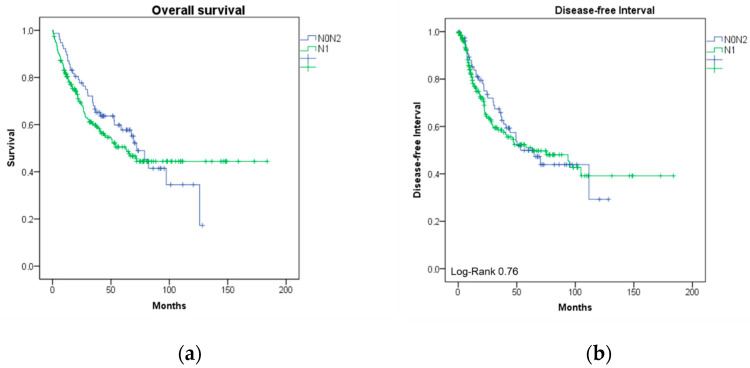
(**a**) Kaplan–Meier plot to overall survival months after surgery; (**b**) Kaplan–Meier plot to disease-free survival months after surgery. Abbreviations: N0N2: skip-N2 metastasis.

**Table 1 jpm-15-00113-t001:** Baseline characteristics: operation and tumor localization.

	Total Cohort	N0N2	N1	*p*-Value
	*n* = 273	*n* = 77	*n* = 196	
Age (years)	64.9 ± 8.8	64.0 ± 9.4	65.2 ± 8.5	0.31
Male gender (%)	67.4	64.9	68.4	0.12
Packyears	38.4 ± 25.9	36.7 ± 25.7	40.7 ± 21.0	0.56
correct preoperative N status via EBUS (%)	47	18	65	0.002
**Comorbidities:**				
One comorbidity (%)	64.5	61.0	65.8	0.56
Multiple comorbidity (%)	31.1	31.2	31.1	0.23
**Histology:**				
AC (%)	72.7	68.8	76.6	0.11
SCC (%)	28.3	31.2	23.4	0.19
**Tumor localization:**				
Left upper lobe (%)	21.6	20.5	23.9	0.45
Left lower lobe (%)	13.2	11.7	13.3	0.12
Right upper lobe (%)	25.6	25.2	25.1	0.99
Middle lobe (%)	6.2	7.4	5.9	0.22
Right lower lobe (%)	12.1	16.9	12.2	0.16
**Operation:**				
Lobectomy (%)	79.8	83.9	79.4	0.34
Bilobectomy (%)	7.7	6.9	8.5	0.45
Pneumonectomy (%)	8.3	8.4	8.2	0.89
Segmentectomy (%)	4.2	5.5	3.9	0.34

Abbreviations: N0N2: skip-N2 metastasis; AC: adenocarcinoma; SCC: squamous cell carcinoma.

**Table 2 jpm-15-00113-t002:** Baseline characteristics: grading and lymph nodes.

	Total Cohort	N0N2	N1	*p*-Value
	*n* = 273	*n* = 77	*n* = 196	
**Grading:**				
G1 (%)	3.3	2.2	2.6	0.78
G2 (%)	50.2	54.2	54.6	0.51
G3 (%)	43.6	45.5	42.9	0.34
Positive lymph-nodes (n)	3.9 ± 4.1	3.8 ± 3.7	4.1 ± 3.6	0.60
Lymph-node ratio (n)	0.19 ± 0.16	0.18 ± 0.12	0.22 ± 0.18	0.11
Tumor size (cm)	3.9 ± 4.1	3.8 ± 3.7	4.1 ± 3.6	0.60
Lymphangosis carcinomatosa (%)	41.0	46.8	38.2	0.51
Hemangosis carcinomatosa (%)	12.8	10.4	13.8	0.54

Abbreviations: G: grading; N0N2: skip-N2 metastasis.

**Table 3 jpm-15-00113-t003:** Overall survival and disease-free interval.

	Total Cohort	N0N2	N1	*p*-Value
	*n* = 273	*n* = 77	*n* = 196	
Disease-free interval (months)	39.5 ± 35.1	40.7 ± 31.1	38.1 ± 36.7	0.45
One-year survival (%)	90.1	90.9	89.6	0.67
Three-year survival (%)	62.2	65.3	60.3	0.20
Five-year survival (%)	42.6	40.7	44.7	0.27

**Table 4 jpm-15-00113-t004:** Cox regression analysis for overall survival.

Variable	HR	CI 95%	*p*-Value
N0N2/N1	1.18	0.79–1.78	0.40
Positive Lymph nodes	1.07	1.04–1.14	0.31
Tumorsize	1.01	0.96–1.08	0.56
Grading	0.99	0.79–1.23	0.91

## Data Availability

The corresponding author will share the article’s data upon reasonable request.

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
