# Peer review of "Non-Small Cell Lung Cancer Patients with Skip-N2 Metastases Have Similar Survival to N1 Patients—A Multicenter Analysis"

_jpm, 2025, doi:10.3390/jpm15030113_

Round 1
Reviewer 1 Report
Comments and Suggestions for Authors
Dear Authors,
Please revise your paper according to my comments below.
My comments:
1. Definitions of skip-N2 ve N1 according to this study should be clearly given in the Methods section.
* Are they defined by surgical results or by PET?
* If defined by PET results, were the surgical results the same as those of PET or different?
2. Results: The N1 and N0N2 groups should be compared regarding the presence of comorbidity, one comorbidity and multiple comorbidites. This should also be added to Table 1.
3. Discussion:
In the paragraph of limitations-
* Please mention about the uncertain cause of death in the study groups (lung cancer or other diseases) and how this might affect survival comparison between N1 and N0N2 groups.
* Also mention about how the adjuvant radiotherapy given to N0N2 group could have affected the survival comparison between N1 and N0N2 groups.
Sincerely,
Reviewer
Author Response
Response to Reviewer 1 Comments
|
||
1. Summary |
|
|
Thank you very much for taking the time to review this manuscript. Please find the detailed responses below and the corresponding revisions/corrections highlighted/in track changes in the re-submitted files.
|
||
2. Point-by-point response to Comments and Suggestions for Authors |
||
Comments 1: 1. Definitions of skip-N2 vs. N1 according to this study should be clearly given in the Methods section. * Are they defined by surgical results or by PET? * If defined by PET results, were the surgical results the same as those of PET or different?
|
||
Response 1: Thank you for pointing this out. We agree with this comment. Therefore, I have inserted a passage in the Methods section. “N0N2 is associated with direct lymphatic drainage into the mediastinal nodes, by-passing the typical spread through pulmonary and hilar lymph nodes. N1 is associated with direct lymphatic drainage into the pulmonary and hilar lymph nodes, only.
page number 3, line 120-122
|
||
Comments 2, 3: Are they defined by surgical results or by PET?
|
||
Response 2, 3: Thank you for pointing this out. But N0N2 and N1 were surgical and consequently pathological results. In order to make clarify this I added a line into the table: “correct preoperative N status via EBUS (%)” Furthermore, I have inserted the following passage in the Results section.
“Preoperative N-status was accurately determined in only 47% of patients. Specifically, N1 was correctly diagnosed in 65% of cases, compared to just 18% for N0 or N2 (p=0.002).” page number 5, section Baseline characteristics, line 203-205
Comments 4: The N1 and N0N2 groups should be compared regarding the presence of comorbidity, one comorbidity and multiple comorbidites. This should also be added to Table 1.
Response 4: Thank you for pointing this out. We agree with this comment. Therefore, I have inserted a passage in the Results section and in table 1. Patients' comorbidities were recorded and classified into two groups: one comorbidity versus multiple comorbidity. No significant differences were observed between these groups (p = 0.56, p = 0.23).
page number 5, section Baseline characteristics, line 206-208
Comment 5: Please mention about the uncertain cause of death in the study groups (lung cancer or other diseases) and how this might affect survival comparison between N1 and N0N2 groups.
Response 5: Thank you for pointing this out. We agree with this comment. Therefore, I have inserted a passage in the limitations section. The survival data may not specify the exact causes of death; however, it is most likely that NSCLC was the primary cause page number 8, section limitations, line 311-312
Comment 6: Also mention about how the adjuvant radiotherapy given to N0N2 group could have affected the survival comparison between N1 and N0N2 groups.
Response 6: Building on the findings of Pechoux et al., it is evident that mediastinal radiation does not offer a survival benefit. Notably, patients with N0N2 disease were treated with radiation as per the guidelines at the time, and combined radiochemotherapy in this group did not result in any significant survival differences.
|
Reviewer 2 Report
Comments and Suggestions for Authors
Dear Authors,
N2 skip metastasis has been lately an active area of research in lung cancer. Although I tend to agree with your overall conclusion that patients with N2 skip metastasis might have a similar prognosis to N1 patients, we need to remember that patients with N2 skip metastasis have been treated more aggressively (radiation therapy in addition to chemo) compared to N1 patients (chemo only). So, this begs the question whether no statistically significant differences between these cohorts was due to more aggressive therapy for patients with N2 skip metastasis. Was there any subset of patients with N2 skip metastasis who are treated similarly to N1 patients? If so, how was the disease-free interval and survival among these groups?
Author Response
1. Summary |
|
|
Thank you very much for taking the time to review this manuscript. Please find the detailed responses below and the corresponding revisions/corrections highlighted/in track changes in the re-submitted files.
|
||
2. Point-by-point response to Comments and Suggestions for Authors |
||
Comments 1: N2 skip metastasis has been lately an active area of research in lung cancer. Although I tend to agree with your overall conclusion that patients with N2 skip metastasis might have a similar prognosis to N1 patients, we need to remember that patients with N2 skip metastasis have been treated more aggressively (radiation therapy in addition to chemo) compared to N1 patients (chemo only). So, this begs the question whether no statistically significant differences between these cohorts was due to more aggressive therapy for patients with N2 skip metastasis. Was there any subset of patients with N2 skip metastasis who are treated similarly to N1 patients? If so, how was the disease-free interval and survival among these groups?
|
||
Response 1: Thank you so much for pointing this out. We have to assume that all patients receive therapy. But in fact, it is not surprising that about 30% are not receiving guideline therapy. Unfortunately, we do not have this date availabl |
Reviewer 3 Report
Comments and Suggestions for Authors
The authors performed a multi-institutional retrospective analysis of overall survival of lung cancer patients found to have pathologic either N1 or skipped N2 (N0N2) nodal metastasis following complete resection and standard of care adjuvant therapy (N1 nodal metastasis ie. stage II: cytotoxic chemotherapy and N2 nodal metastasis ie. stage III: chemotherapy and thoracic radiation). the authors demonstrated that these patient populations had comparable 1-3- and 5-year overall survival. I have the following comments
- why did the authors not include in this analysis of patients with N1N2 metastasis to demonstrate that N1 and N0N2 patients have similar survival outcome while N1N2 patients presumably had worse outcome. this will strengthen the argument that N1 and N0N2 have similar oncologic behavior.
- While I cannot refute the finding that the overall survival of the two groups are identical, I cant quite agree that they have similar prognosis. they were treated differently in the adjuvant setting. If the N0N2 group was treated with chemotherapy alone and no radiation therapy then I can "buy" that argument.
Author Response
1. Summary |
|
|
Thank you very much for taking the time to review this manuscript. Please find the detailed responses below and the corresponding revisions/corrections highlighted/in track changes in the re-submitted files.
|
||
2. Point-by-point response to Comments and Suggestions for Authors |
||
Comments 1: The authors performed a multi-institutional retrospective analysis of overall survival of lung cancer patients found to have pathologic either N1 or skipped N2 (N0N2) nodal metastasis following complete resection and standard of care adjuvant therapy (N1 nodal metastasis ie. stage II: cytotoxic chemotherapy and N2 nodal metastasis ie. stage III: chemotherapy and thoracic radiation). the authors demonstrated that these patient populations had comparable 1-3- and 5-year overall survival. I have the following comments why did the authors not include in this analysis of patients with N1N2 metastasis to demonstrate that N1 and N0N2 patients have similar survival outcome while N1N2 patients presumably had worse outcome. this will strengthen the argument that N1 and N0N2 have similar oncologic behavior.
Response 1: Thank you so much for pointing this out. This question is straightforward to answer, as our research group has already addressed it in a previous publication—with me as first author—comparing N0N2 and N1N2 patients. The results were unequivocal: long-term survival was significantly worse for N1N2 patients compared to those with N0N2 disease.
https://pubmed.ncbi.nlm.nih.gov/37369037/
Comment 2: While I cannot refute the finding that the overall survival of the two groups are identical, I cant quite agree that they have similar prognosis. they were treated differently in the adjuvant setting. If the N0N2 group was treated with chemotherapy alone and no radiation therapy then I can "buy" that argument.
Response 2: Thank you so much for pointing this out. They were treated differently due to the guidelines given. Building on the findings of Pechoux et al., it is evident that mediastinal radiation does not offer a survival benefit. In this respect, this comparison will be interesting again in about 5 years when we have long-term data on these two groups, as they now received the same adjuvant therapy.
|
||